# TRI-POSE-Net: Adaptive 3D human pose estimation through selective kernel networks and self-supervision with trifocal tensors

**Nabeel Ahmed Khan**[1], **Aisha Ahmed Alarfaj**[2], **Ebtisam Abdullah Alabdulqader**[3], **Nuha Zamzami**[4], **Muhammad Umer**[5], **Nisreen Innab**[6]*, **Tai-Hoon Kim**[7]*

1 Center For AI and Big Data, Namal University, Mianwali, Pakistan, 2 Department of Information Systems, College of Computer and Information Sciences, Princess Nourah bint Abdulrahman University, Riyadh, Saudi Arabia, 3 Department of Information Technology, College of Computer and Information Sciences, King Saud University, Riyadh, Saudi Arabia, 4 Department of Computer Science and Artificial Intelligence, College of Computer Science and Engineering, University of Jeddah, Jeddah, Saudi Arabia, 5 Department of Computer Science & Information Technology, The Islamia University of Bahawalpur, Bahawalpur, Pakistan, 6 Department of Computer Science and Information Systems, College of Applied Sciences, AlMaarefa University, Diriyah, Riyadh, Saudi Arabia, 7 School of Electrical and Computer Engineering, Yeosu Campus, Chonnam National University, Yeosu-si, Jeollanam-do, Republic of Korea

* Ninnab@um.edu.sa (NI); taihoonn@chonnam.ac.kr (THK)

**Data Availability Statement:** The datasets utilized in this research is publicly available at these links and can also be requested from the authors. First

## Abstract

Accurate and flexible 3D pose estimation for virtual entities is a strenuous task in computer vision applications. Conventional methods struggle to capture realistic movements; thus, creative solutions that can handle the complexities of genuine avatar interactions in dynamic virtual environments are imperative. In order to tackle the problem of precise 3D pose estimation, this work introduces TRI-POSE-Net, a model intended for scenarios with limited supervision. The proposed technique, which is based on ResNet-50 and includes integrated Selective Kernel Network (SKNet) blocks, has proven to be efficient for feature extraction customised specifically to pose estimation scenarios. Furthermore, trifocal tensors and their trio-view geometry allow us to generate 3D ground truth poses from 2D poses, resulting in more refined triangulations. Through the proposed approach, the 3D poses can be estimated from a single 2D RGB image. Moreover, the proposed approach was evaluated on the HumanEva-I dataset yielding a Mean-Per-Joint-Position-Error (MPJPE) of 47.6 under self-supervision and an MPJPE of 29.9 under full supervision. In comparison with the other works, the proposed work has performed well in the self-supervision paradigm.

## 1 Introduction

In the framework of the developing human 3d pose estimation, the goal of endowing machines with vision capabilities akin to those of humans is becoming increasingly prominent. Advances in computer vision and human-computer interaction illustrate the importance of accurately perceiving and interpreting the behaviors of virtual entities. Human pose estimation was first introduced in response to [1], which fits in well with the virtual environment's need

HumanEva-I dataset direct link (http://humaneva.is.tue.mpg.de/datasets_human_1). Second MPII Human Pose dataset direct link (http://human-pose.mpi-inf.mpg.de/#download).

**Funding:** This study was supported by Princess Nourah Bint Abdulrahman University Researchers Supporting Project number (PNURSP2024R348), Princess Nourah Bint Abdulrahman University, Riyadh, Saudi Arabia. Nisreen Innab would like to express sincere gratitude to AlMaarefa University, Riyadh, Saudi Arabia, for supporting this research.

**Competing interests:** The authors have declared that no competing interests exist.

for realistic avatars and entities that can move around and interact with virtual environments in a believable way. Similar to its practical uses in sports science, rehabilitative training, video surveillance, and behavioral identification, human posture estimation plays a significant role in the in human 3d pose estimation [2, 3]. Although conventional 3D human pose estimation usually uses depth data from specialist devices such as Microsoft Kinect, the realistic environment requires more easily available options. The usefulness and affordability of using standard RGB cameras become clear in this situation. The estimation of virtual human positions based on regular RGB pictures is a study area that is receiving significant attention and practical significance in the virtual environment and beyond. Both industry and academics are actively investigating this topic. This change is in line with easily accessible and immersive virtual experiences, as well as addressing the drawbacks and high expenses of conventional methods. Two broad categories of deep learning algorithms and traditional schemes can classify the most commonly used 3D human pose estimators. Conventional methods, used early in the progress, were based on a deformable part model and a body part detector with graph structure. These techniques employed a graph model with scripted features, such as HOG and SIFT, to described the spatial relationships [1, 4]. But, these methods had their limitations because they were dependent of the appearance of the image and failed to fully exploit image features.

A number of artificial neural networks are now employed in identification and attributes estimation in computer vision's big data age. The Human posture estimation research has shifted from classical to deep learning approaches, whereby pre-dominnantly CNN were used to learn humans pose data directly from images. Works such as [5, 6] revealed how perfectly deep neural networks perform in order to predict human location in a three-dimensional space from a single picture. Even with the impressive successes of current approaches, there are still significant obstacles to overcome in order to deploy 3D human pose estimation at the application level. The commonly utilised 3D datasets, which are single-image backgrounds and 3D ground truth labels collected by motion capture systems (MOCAPs) in controlled environments, are expensive. Models trained on such datasets confront difficulty in extending to outside and complicated situations with body part occlusion. In this paper, we address this problem by proposing a network model for 3D human pose estimation that uses a self-supervised training approach to reliably estimate 3D human posture from photos lacking 3D labels.

The name 'Tri-POSE-Net' reflects the core principles of the model: the use of trifocal tensors, trio-view geometry, and three modules added into the architecture—ResNet-50, SKNet blocks, and trifocal tensors for proper 3D pose computation. The proposed technique uses ResNet-50 for feature learning while the SKNet blocks available dynamically adjust to pose estimation scenario using different scales and orientations. The trifocal tensors are important because they help in generating ground truth poses from 2D poses across three views resulting to much accurate triangulations. In this manner, TRI-POSE-Net efficiently estimates 3D poses from a single 2D RGB image while the structure of this work includes the threefold strategy. All abbreviations utilized in this research work are shared in Table 1. The major contributions of this paper are listed below.

1. A model named TRI-POSE-Net is proposed under the weakly-supervised 3D pose estimation paradigm. The model consists of a ResNet-50 baseline modified using two SKNet blocks, introduced with the intent of dynamically allocating the receptive field sizes and the global context based adaptive feature selection for a refined pose estimation.

2. Transfer learning is thoroughly implemented as part of the first training regimen for the acquisition of 2D poses, by training the 2D variant of the proposed TRI-POSE-Net on in-the-wild dataset. This parameter sharing amplifies the generalization capabilities of the model whilst generating 2D poses on HumanEva-I dataset.

**Table 1. Acronym table.**

| Abbreviation | Description |
|---|---|
| 3D | 3-Dimensions |
| SKNet | Selective Kernel Network |
| MPJPE | Mean Per Joint Position Error |
| CNN | Convolutional Neural Network |
| RGB | Red Green Blue |
| HOG | Histogram of Oriented Gradients |
| SIFT | Scale-Invariant Feature Transform |
| LiDAR | Light Detection and Ranging |
| MRM | Model Recursive Matching |
| MOCAP | Motion Capture |
| 2D | 2-Dimensions |
| LBP | Local Binary Patterns |
| RF | Random Forest |
| RESNeT | Residual Network |
| SGD | Stochastic Gradient Descent |
| SVM | Support Vector Machine |
| VGG | Visual Geometry Group |

3. The use of trifocal tensors in this context primarily aids in capturing global spatial relationships. By establishing geometric constraints and correspondences between three different views, trifocal tensors provide a more comprehensive understanding of the overall 3D structure, which goes beyond the local spatial details. This global spatial information is crucial for accurately reconstructing 3D poses from multiple viewpoints.

## 2 Related works

This section describes some state-of-the-art closely related research works related to the topic of human 3D pose estimation. We have utilized trifocal tensors for weakly and self-supervised 3D human pose estimation. We have categorized related work into two subsections to separate the approaches.

Work has progressively focused on developing techniques in the relevant field of study for 3D human pose estimation primarily concentrating on modality oriented methods. To this end, Lee et al. [7] proposed, in their pioneering study, an MCMC method specifically designed for approximating the 3D HP of the human upper body. This allowed to show how the proposed method works and that it can be successfully used for images of different nature, which proved the applicability of the new approach. Examining further the progressive aspect of modalities, Jung et al. [8] developed a new approach to the 3D human pose estimation, with reference to depth images. One of their methods reconceptualised the problem as a detection of desired body joints, which also demonstrated a proper appreciation of the complex processes required to obtain depth-based human poses. Based on this, Wu et al extended the work in the area of single image based human pose estimation with the use of depth maps and they proposed the model-based recursive matching (MRM) technique. This method placed the field on a different level of professionalism and was an input to the discussion on the existing techniques of 3D human pose estimation.

To this end, Lu et al. (2024) introduce the *AvatarPose*, a new approach to estimate the 3D pose in the context of proximal interactions from inadequate multiview videography. This approach uses the prior Personalized Implicit Neural Avatars, which are beneficial to enhance the robustness and the correspondence's accuracy of body pose estimation; especially when there are occluded or contacted areas. The avatars are reconstructed with the help of layered volume rendering, it permits direct optimization of 3D poses according to the color and silhouette rendering losses. This method is superior to other models in datasets with tightly correlated objects with each other [9, 10]. In this work, the authors introduce *PoseResNet* for self-supervised learning of 3D human pose, without the use of 3D ground truth labels. The model adopts ResNet-50 as its base model and adopts CBAM as well as its proposed multi-scale feature fusion module named WASP. A deconvolution network is then employed to generate volume heatmaps which are then passed through a soft argmax function to estimate joint coordinates. This method proves better performance where, with an average of calculating per joint position error (MPJPE), it was 74. 6 mm with removing the need for 3D labels [11]. To the best of the authors' knowledge, Zhou et al. proposed *Trifocal PoseNet* in 2024 as a model targeting at learning 3D poses with a small amount of annotations. This model uses trifocal tensors to make good use of the geometrical constraints and the relationships between the three views to improve the triangulation. Selective kernel network is proposed to employ within ResNet-50 framework, in order to capture the features of pose and optimize them. This presented approach shows how trifocal geometry can be used to synthesize poses from 2D images even with little requirement for supervision [12].

Similarly, another set of works strictly bounded to the realm of 3D human pose estimation has focused on extending the applicability of these techniques in particular application areas in order to demonstrate their efficacy. Lie et al. [13] switch their focus on research to the important area of elderly health monitoring. Their proposal was to extract 3D skeletons as a way of surveillance of daily living among the elderly. This application-oriented approach is in line with the flexibility and the practicality of the 3D human pose estimation. Adding to the list of application domains of AE, Moon et al. [14] put forward a human pose refinement network for multi-person pose estimation. This work while being theoretical also dealt with issues to do with computation costs and elimination of pipeline effects, that are relevant in the real world application of these technologies. Moreover, Cong et al. [15] indicated a method that uses monocular camera and single LiDAR for 3D multi-person pose estimation in large-scale scenes. Stressing more on the indispensability of depth information, this work ensured that 3D pose estimation skills are viable in all real world settings hence not limited to simple settings. Altogether, the works considered herein form a wide survey of 3D human pose estimation with focus on the progressive modalities and a high level of practical concern, which proves the potential and applicability of the techniques.

## 2.1 Weakly and self-supervised approaches in 3D human pose estimation

A number of works are devoted to weakly and self-supervised methods for 3D human pose estimation. We also refer to papers [16–18], where authors studied how to use weakness and self-supervision for overcoming some difficulties connected with receipt of 3D ground truth labels. In a way, self-supervised and weakly supervised methods sketched in this paper provide a partial solution to this problem that the fully supervised approach [19–21] addressing direct estimation of 3D joint coordinates from images using regression networks. Pavlakos et al. [22] utilized a pictorial structure model to derive global pose configurations from key-point heat maps in multi-view images. However, their method required full camera calibration and a key-point detector for generating 2D heat maps. Rhodin et al. [16] used multi-view consistency

restraints to guide a network, minimizing the need for extensive 3D ground truth data to avoid pose collapse issues. Nonetheless, the scarcity of in-the-wild 3D ground truth data remains a limitation. Drover et al. [17] introduced an adversarial framework for weak supervision, learning 3D structures solely from random 2D projections. Notably, their method eliminated the need for correspondence between 2D and 3D points but still relied on 3D labels. Zhou et al. [23] integrated 2D and 3D data, applying a Euclidean loss for 3D data and proposing a weak supervision loss based on 2D annotations and prior knowledge of the human skeleton. Despite improving applicability to outdoor images, their approach required both 2D and 3D labels. Wandt et al. [24] pioneered a generative adversarial network with three components, including a pose and camera estimation network, a critic network, and a re-projection network. Their method was also accompanied by another feature: camera estimation network and re-projection layer for re-projection of the estimated 3D pose to 2D space.

Rhodin et al. [25] proposed a weakly supervised method to transfer the body representation from the 3D space that is adapted to the geometric analyses of the multiple-view images. In their Self-supervised learning paper, Kocabas et al. [26] introduced Epipolar-Pose which doesn't require 3D ground truth data or cameras extrinsics information instead it uses epipolar geometry to produce annotations from key-points. A self-supervised method of 3D human pose estimation from monocular images was presented by Li et al. [27, 28]. Altogether, these works help the investigation of the methods that need little control and supervision, which respond to issues related to the annotated data availability. Extending the weak supervision topic further, Wu et al. [29] introduced a work on weakly supervised 3D human pose estimation using the transfer of 2D HPE annotation information from large scale RGB datasets to the 3D task. Roy et al. [30] discussed a supervised method making use of multi-view geometrical restraints which can be considered as a form of self-supervision where the labelled data is limited. Cha et al. [31] proposed a new self-supervised learning paradigm for 3D human pose and shape recovery in single shot 2D images. Altogether, these works are connected to the investigation of the self-supervised learning frameworks and present progress in handling supervision issues in 3D pose estimation.

## 3 Methodology

In the method described in this study, self-supervision is used to train a CNN for the purpose of 3D human pose estimation. It is important to underline that this training process subdivides into two different stages. The first of these is a basic 2D human pose estimation network which is kept fixed during the training process. The key idea of this module is that of convolutional networks which are optimized for the purpose of producing 2D pose estimates from images, and is built using a variant of ResNet-50 architecture. The second training model involved in the training pipeline is a 3d pose estimation model. Similar to the previous model, this is a derivation of the modified ResNet-50 architecture and goes through training in a weakly supervised manner. The training leverages 3D-pose ground truths, which are derived by post processing the 2D-poses through epipolar geometry techniques.

### 3.1 TRIPOSE-NET

The first baseline network adopted in the TRIPOSE-NET module is ResNet-50 due to its ability to mitigate on vanishing gradients which is a common problem in deep convolutional network. The nature of these vanishing gradients is that they reduce the effectiveness of convolutional networks moving to the deeper layers while it seems that more layers should be better at capturing finer details. The possibility of degradation in performance especially in deep networks is also prevented by residual connections which enable the gradients to flow

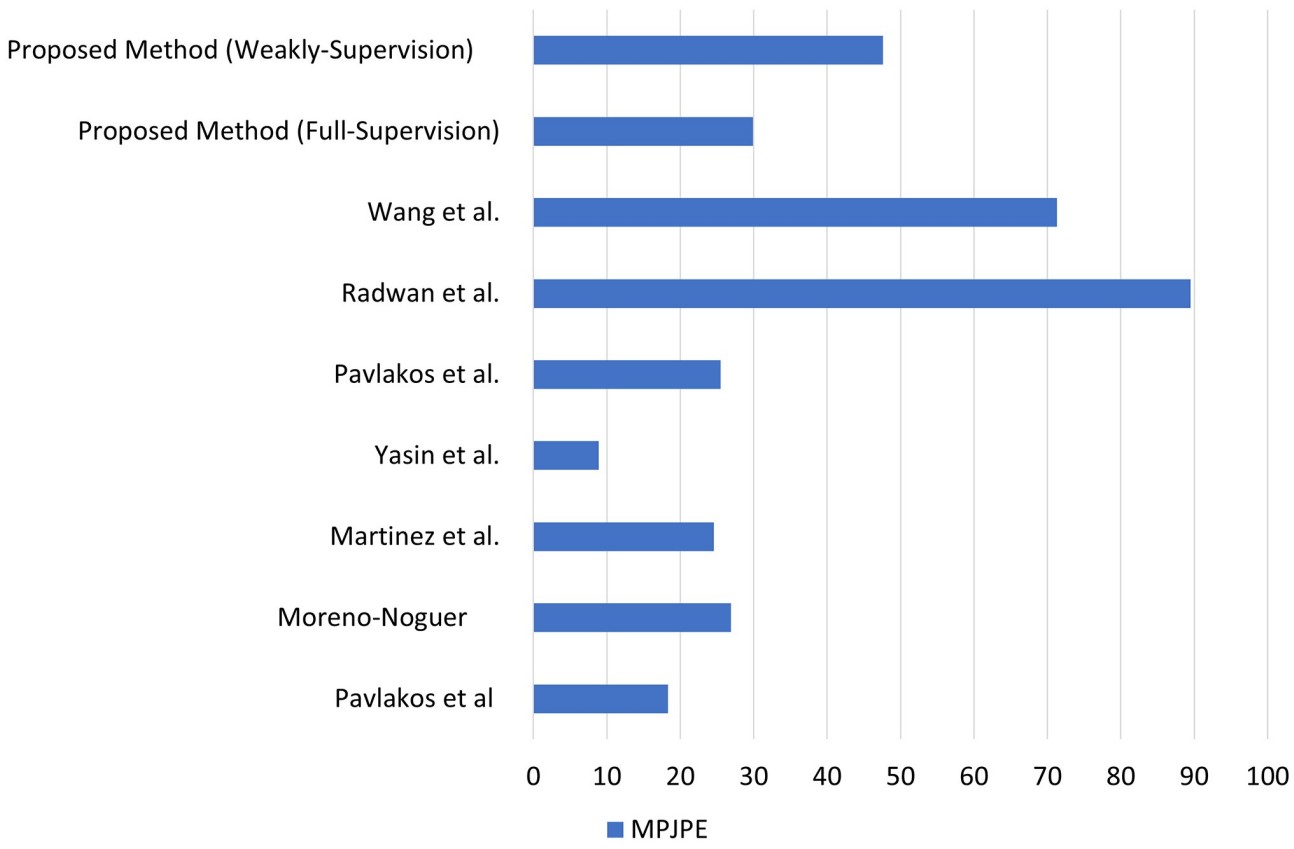

**Fig 1. Modified feature extraction block (ResNet-50) using selective kernel network.**

through the depth of the network thereby making the optimisation possible. Owing to its versatility across a plethora of tasks, the Residual Network (ResNet) is an optimal choice as the backbone for these pose estimation pipelines, providing liberty to extract abstract and densely intricate features by virtue of its deep and robust distinctive features. To elevate the performance of the ResNet-50 module, an SKNet Block is introduced at the end of the 3rd convolution block and at the tail-end of the feature extractor, i.e., just before the global average pooling layer.

This forms an effective feature extraction module named SK-ResNet 50, as illustrated in Fig 1. The addition of the SKNet block at the deeper layers, primarily the bottleneck blocks, assures that more global context is captured as the receptive field is larger in this region and that the layers have a high spatial resolution. Additionally, the placement of the SKNet block before the global average pooling allows for adaptive feature selection on the basis of global context prior to the final prediction. Furthermore, using SKNet in ResNet-50 augments the model's capabilities in dealing with multi-scale features by acting as an auxiliary aid in selecting the appropriate kernel sizes for capturing both fine-grained and global details to achieve optimal performance in pose estimation.

Initially, a deconvolution layer is used to project the extracted feature from the 2D pose estimation module as Heat-Maps denoted by $M_h$, where $M_h \in \mathbb{R}^{m \times n}$. Here $(m, n)$ represent the dimensions of $M_h$ and to acquire the 2D key-point positions $(x, y)$ from the respective maps $M_h$ for all nodes $N$ such that $A \in \mathbb{R}^{N \times (x,y)}$, we apply an argmax function $v$.

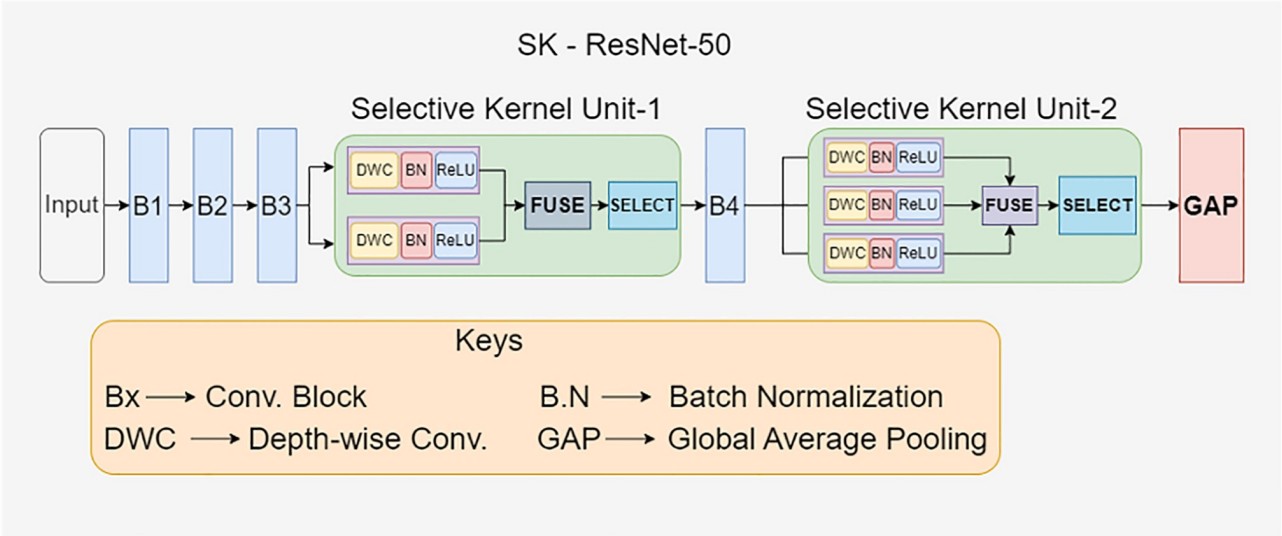

**Fig 2. Layer topology of selective kernel unit.**

The resultant human poses $A_{(i,\,j)}$ are in two-dimensional representation, whereas the acquisition of 3D poses requires supervision under a 3-dimensional paradigm. The annotation of 3D poses for training endeavors is both time-consuming and resource-intensive. Herein, we propose a weakly supervised method, utilizing the 2D poses $A_{(i,\,j)}$ from initial training and epipolar geometry to triangulate the 3D correspondences. The volumetric key-point mappings given by $C \in \mathbb{R}^{N \times (x,y,z)}$ are cached and taken as ground truths with pseudo labels $\tilde{L}$ to train the TRIPOSE-NET tailored to 3D pose estimations $C_{(i,\,j,\,k)}$, significantly curbing the need for pre-annotation of the data for 3D pose estimates.

## 3.2 Selective kernel network

Selective Kernel Network (SKNet) [32] emulates the dynamic modulation of receptive fields of visual-cortical neurons based on stimuli. This emulation allowed the extension of CNN with the integration of adaptive kernel sizing based on the input information. In SKNet, a Selective Kernel (SK) unit containing multiple branches with varying kernel sizes is fused in a way that ascertains the appropriate selection of the receptive field size, as visually explained in Fig 2. This adaptability equips the model with the ability to focus only on the most informative regions, akin to an attention mechanism. With proficient multi-scale feature integration owing to the dynamic allocation of receptive field size, SKNet extracts both coarse-level and fine-grained features imperative for effective pose estimation [33]. Additionally, the selective kernel feature provides a more expressive feature representation, with different parts of the network specializing in capturing diverse features. Relationships between features are meaningfully captured, resulting in improved pose estimation. A task where the preservation of the relationship between the joints is a significant facet. Within the proposed TRIPOSE-NET architecture, one of the SK units is added before the 4th convolution block of ResNet-50, and a second SK unit is added before the Global-Average-Pooling Layer, as shown in Fig 1. The 1st SK unit is dual-branched with two kernels of sizes 3x3 and 5x5. The 5x5 kernel is actually a dilated version of a 3x3 kernel with a dilation factor of 2. Dilated convolutions [34], as shown in Fig 3, help capture contextual information by introducing gaps within the elements of the filter, effectively reducing the parameter count and enhancing the network's global awareness.

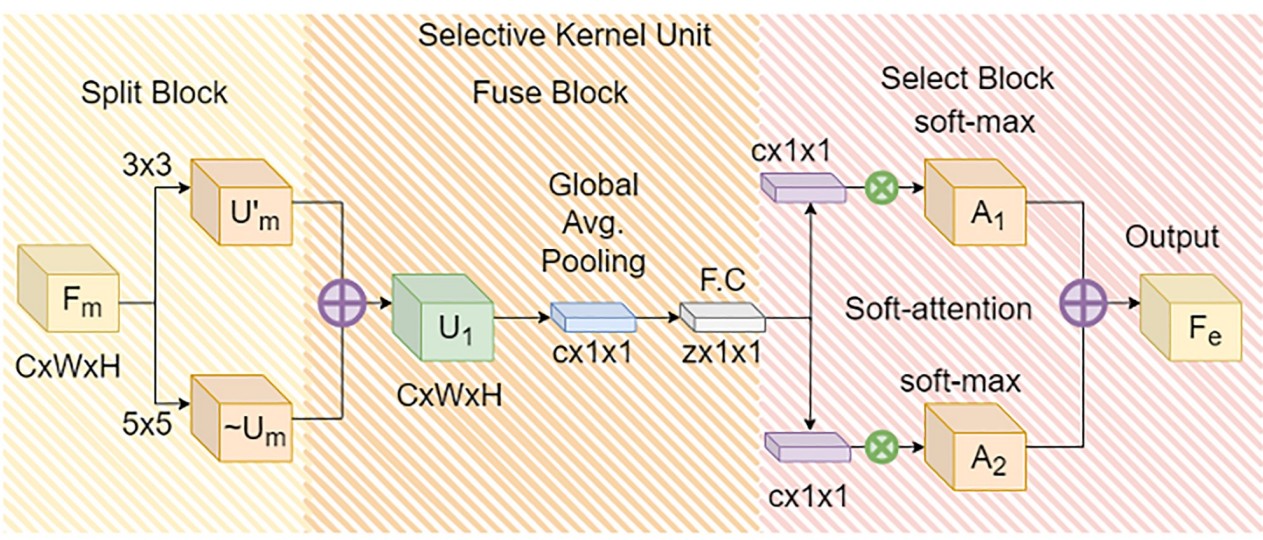

**Fig 3. Visual illustration of dilated convolution kernel and its receptive field.**

The 2nd SK unit consists of three branches with kernel sizes of 3x3, 5x5, and 7x7. The first kernel is without dilation, while the other two are 3x3 kernels dilated with factors 2 and 3.

The SK unit is segregated into three operations, namely Split, Fuse, and Select. Within the Split operation, the feature map $F_m$ is propagated to the SK module, where two transformations $\tilde{F}_m$ and $\hat{F}_m$ are performed. Each transformation involves a depth-wise convolution, followed by batch normalization, and finally a ReLU activation, in this sequence. For the first integration of the SK block within the ResNet-50 module, a dual-branch transformation $\tilde{F}_m$ and $\hat{F}_m$ is used with kernel sizes 3x3 and 5x5, respectively. While the second embedding of the SK unit is based on a three-branched transformation $\tilde{F}_m$, $\hat{F}_m$, and $\acute{F}_m$, each branch with a varying receptive field size as shown in Fig 2. The respective transformations within the Split operation produce outputs $\tilde{U}_m$, $\hat{U}_m$, $U_m^n$, where $n$ represents the number of transformations.

The Fuse operation involves the element-wise summation of each transformation, which in our case involves both the two-branched $U_1$ and three-branched $U_2$ transformations, given by:

$$U_1 = \tilde{U}_m + \hat{U}_m \tag{1}$$

$$U_2 = \tilde{U}_m + \hat{U}_m + \acute{U}_m \tag{2}$$

The outputs $U_1$ and $U_2$ from the fused transformations are passed through a Global Average Pooling Operation to obtain channel-wise global information statistics. The Pooling Operation $G_p$ is applied across the spatial dimensions $W \times H$ for each element $e$ of the output vector $P \in \mathbb{R}^e$. The pooling operation is given as

$$P_e = G_p(U_e) = \frac{1}{H \times W} \sum_{i=1}^{H} \sum_{j=1}^{W} U_e(i, j) \tag{3}$$

The output vector $P_e$ is further compressed to obtain a compact vector $c$, by reducing the dimensionality via a fully connected ($fc$) layer, ensuring optimal efficiency in feature selection.

$$c = fc(P) = \phi(\omega(K \cdot P)) \tag{4}$$

In the above equation, $\varphi$ represents the ReLU activation, $\omega$ represents Batch Normalization, and $K \in \mathbb{R}^{d \times e}$, where $d$ is the depth of the vector.

Similarly, the Select operation selects different spatial scale information by the application of soft attention across the channels, guided by the compact vector $c$. Softmax is applied channel-wise such that for a two-branched network $U_1$, the mathematical operation to extract the attention score is given by

$$a_e = \frac{e^{A_{ec}}}{e^{A_{ec}} + e^{B_{ec}}} \, , \quad b_e = \frac{e^{B_{ec}}}{e^{A_{ec}} + e^{B_{ec}}} \tag{5}$$

The final feature map $F_e$ is given by:

$$F_e = a_e \cdot \tilde{U}_m + b_e \cdot \hat{U}_m \tag{6}$$

where,

$$a_e + b_e = 1 \tag{7}$$

The above Eq (7) can be extended for a three-branched network $U_2$, used in this study, by calculating $c_e$ through the third matrix $C$, acquiring the final feature map $F'_e$ of $U_2$ as

$$F'_e = a_e \cdot \tilde{U}_m + b_e \cdot \hat{U}_m + c_e \cdot \acute{U}_m \tag{8}$$

## 3.3 Dataset description

Existing 3D pose estimation datasets face a myriad of challenges, including limited diversity, ambiguous depth estimates, training data limitations, and predominantly indoor and controlled setups. These limitations impede progress in advancing 3D pose estimation algorithms to tackle the complexities of real-world scenarios. To somewhat bridge the gap in environmental diversity, this study utilizes two datasets, namely MPII and HumanEva-I, to familiarize the 3D pose estimation model with outdoor conditions. The MPII 2D Human Pose dataset [35] is widely used in computer vision due to its diverse and "in-the-wild" nature. This dataset contains around 25,000 photos capturing a wide range of human poses in different real-world scenarios. Each image is meticulously annotated with 16 different body joints, covering important anatomical regions.

The HumanEva-I dataset [36, 37] was created to evaluate and assess the performance of 3D pose estimation algorithms. It focuses on scenarios involving dynamic movements of multiple subjects and is annotated with 3D ground truth keypoints for each frame. It includes 15 anatomically significant joints such as the pelvis, hips, ankles, and thorax. The dataset includes seven video sequences—four in grayscale and three in color—featuring four annotated individuals engaged in five distinct actions. The MPII dataset is used for the initial training of the 2D pose estimation module, while HumanEva-I is employed for weakly supervised training of the 3D pose estimation pipeline. The MPII dataset's skeleton topology contains 16 annotated joints, whereas HumanEva-I's skeleton topology includes 15. Therefore, the neck joint is omitted during training for consistency.

## 3.4 Supervised pre-training on MPII

In this study, the first training phase of the entire 3D pose estimation pipeline involved using the MPII dataset to train the baseline 2D pose estimation model in a supervised setting. Pre-training on the MPII dataset aims to achieve optimal performance in realizing 2D poses in

diverse outdoor environments, as these trained weights are later leveraged during the training of the 2D model on the HumanEva-I dataset. By increasing the generalization ability of the baseline 2D pose estimation model through pre-training on MPII, the second training phase on the HumanEva-I dataset, using parametric transfer, yields improved results, considering the indoor nature of the dataset and its MOCAP-based acquisition. High pose estimation accuracy from the 2D network is crucial for efficient self-supervised training of the 3D pose estimation model. The pose estimates generated by the 2D model are triangulated to obtain the respective 3D point estimates and used as cached ground truths for training the 3D pose estimation model.

## 3.5 Self-supervised training

Weak or Self-Supervised training refers to a training regimen where the data is not explicitly pre-annotated through manual labor; instead, alternative methods are employed for automated ground truth generation. All of them normally rely on some additional algorithms or properties of training data sets or their structures. While in simple and computational inexpensive tasks like age estimation, pose estimation it is pretty reasonable, but in sophisticated and time and computation consuming tasks such as 3D pose estimation, one depends on 3D annotated data and it forms bottleneck. Self-supervised methods give an efficient way to minimize the reliance on human inferences, which in the long run eliminates costs and manual efforts. In the current paper, we show a way to obtain the ground truth 3D poses for training the 3D pose estimation model in an unsupervised manner by utilizing trifocal tensors, which is explained in the subsequent section.

There are three views considered in giving unsupervised 3D ground truths required in training the 3D pose estimation module. The selection of three views is due to trifocal tensors which are used in the current study as a contrast to the fundamental matrix approach where two-view geometry is used. The trifocal tensors provide control of correspondence measurements as three-view geometry proposed by the trio-view approach, and, therefore, make the estimates of 3D pose more accurate. Also, problems of uncertainty in pose estimates of the objects are reduced inasmuch as stringencies inherent in the restraints make the estimate less vulnerable to degenerate solutions. To train the upper part of the pipeline, a trio of images $L_1$, $L_2$, and $L_3$ with a resolution of $728 \times 728$ from the HumanEva-I dataset are fed into both the frozen 2D pose estimation and the 3D pose estimation modules. The output pose estimates $A_1$ $_{(x, y)}$, $A_{2(x, y)}$, and $A_{3(x, y)}$ from each trio are geometrically transformed using the trifocal tensor-based estimate to generate the 3D vector $\hat{C}$ within the confines of the global coordinate frame. Similarly, the 3D pose estimation model outputs a vector $\hat{K}$ containing the estimations for the joints in 3D space.

Under the pinhole camera projection model, let the $p^{th}$ joint in the $i^{th}$ image represent the 2D coordinates given by $R_{(i, p)} = [x_{(i, p)}, y_{(i, p)}]$, while the respective 3D coordinates are given as $Q_p = [X_p, Y_p, Z_p]$. The relation of 3D points in homogeneous coordinates within the image plane for the pinhole projection model can be given by:

$$\begin{bmatrix} x_{(i,p)} \\ y_{(i,p)} \\ \omega_{(i,p)} \end{bmatrix} = K[R|\mathbf{T}] \begin{bmatrix} X_p \\ Y_p \\ Z_p \\ 1 \end{bmatrix} \tag{9}$$

Here, $(x_{(i, p)}, y_{(i, p)}, \omega_{(i, p)})$ represent the homogeneous coordinates on the image plane while

$(X_p, Y_p, Z_p)$ are the 3D points in the camera coordinate system. $T$ and $R$ are the extrinsic parameters that give the translation and rotation of the camera while $K$ holds the intrinsic parameters of the camera inclusive of focal lengths $f_x, f_y$ and the principal points $c_x, c_y$ and is mathematically expressed as

$$K = \begin{bmatrix} f_x & 0 & c_x \\ 0 & f_y & c_y \\ 0 & 0 & 1 \end{bmatrix}, \quad T = \begin{bmatrix} T_x \\ T_y \\ T_z \end{bmatrix} \tag{10}$$

From the above definitions, the fundamental matrices $F_{12}, F_{23}, F_{31}$ for each pair of views, i.e., $[1, 2], [2, 3], [3, 1]$ in a trio-view geometry, can be calculated as under

$$F_{ip} = K_i^{-\top} T K_p^{-1} \tag{11}$$

The epipolar restraints of 2D coordinates for the trio-view can be defined as

$$R_{2,p}^\top F_{12} R_{1,p} = 0 \tag{12}$$

$$R_{3,p}^\top F_{23} R_{2,p} = 0 \tag{13}$$

$$R_{1,p}^\top F_{13} R_{3,p} = 0 \tag{14}$$

Using the trifocal tensor $T_f$ to express the fundamental matrices, we get

$$T_f = [F_{23}]^\top F_{12} + F_{31} \tag{15}$$

Given the points $(R_{1,p}, R_{2,p})$, the third epipolar line $l_{3,p}$ can be computed using

$$l_{3,p} = T_f \begin{bmatrix} R_{1,p} \\ R_{2,p} \\ 1 \end{bmatrix} \tag{16}$$

The 3D position of the joint $R_{3,p}$ can now be triangulated by computing the point of intersection with respect to the joint in each view for all three epipolar lines. The output 3D coordinates for each point are used to supervise the training of the 3D Pose estimation model. During self-supervised training, the 3D pose estimation model predicts the 3D coordinates from the input image for each of the corresponding joints given by $J_{(x, y, z)}$, the loss is then calculated between the cached ground truths $\hat{C}_{(x,y,z)}$ estimated using epipolar geometry and the predicted 3D key-points $J_{(x, y, z)}$. By projecting the predicted 3D key-points $J_{(x, y, z)}$ into the camera space, the loss $L_j$ can be computed by minimization of $\text{smooth}_{L1}(J_{(x,y,z)} - \hat{C}_{(x,y,z)})$ by substitution of $x = (J_{(x,y,z)} - \hat{C}_{(x,y,z)})$.

$$\text{smooth}_{L1}(x) = \begin{cases} 0.5x^2 & \text{if } |x| < 1 \\ |x| - 0.5 & \text{otherwise} \end{cases} \tag{17}$$

## 4 Results and discussion

This section discusses about different types of results we obtained in this research work utilizing different learning models on this dataset. This section also contains a detailed comparison with previously published research works.

### 4.1 Evaluation metrics

In this subsection, we have given a brief introduction of all evaluation metrics we have utilized in this research work.

**4.1.1 MPJPE.** The Mean Per Joint Position Error or MPJPE is a common measure of the performance of 3D pose estimation methods that is used to determine the level of precision. MPJPE aims at computing the mean Euclidean distance between the re-projected and ground truth 3D locations of a joint. This is as simple as a single scalar value; the average error over all joints and frames as has been estimated by MPJPE. Here are the results of the performance of 3D pose estimation and how the MPJPE affects it. Also, it is as interesting to realize that MPJPE is scale-invariant of 3D positions, which means that it depends on the scale [38, 39]. Normalization may be used where the error is expressed as a portion of some reference size of the subject, such as area. The mathematical expression for MPJPE is as follows:The mathematical expression for MPJPE is as follows:

$$\text{MPJPE} = \frac{1}{F \cdot K} \sum_{i=1}^{F} \sum_{j=1}^{K} \parallel G_{i,j} - A_{i,j} \parallel \tag{18}$$

Here,

- $F$ = Number of Frames.

- $K$ = Number of Key-points.

- $G_{i,j}$ = The 3D position of key-point $j$ in frame $i$.

- $A_{i,j}$ = The Predicted 3D location of joint $j$ in frame $i$.

The term $\parallel G_{i,j} - A_{i,j} \parallel$ reflects the Euclidean distance between the projected 3D coordinates of joint $j$ in frame $i$ and its ground truth. These variables are involved in the computation of MPJPE because they quantify the mean Euclidean distance between the estimated and ground truth joint locations across frames and joints giving information about the accuracy of 3D pose estimation methods.

The Mean Per Joint Position Error (MPJPE) is most preferred for the evaluation of human 3D pose estimation because of its quality of directly measuring the precision of estimated poses by using the raw Euclidean distance in between corresponding joints in the predicted and ground truth poses [40]. This assessment is at the joint level, which gives detailed information on algorithm performance region by region to allow researchers to locate specific problems. Furthermore, MPJPE is reliable concerning pose variability and straightforward to analyze, which has led to its wide use as a benchmarking tool in the field and cross-study comparability. Due to its coverage and general appraisal as a successful measure, MPJPE serves as a stable and successful measure of human 3D pose estimation.

**4.1.2 Percentage of Correct Keypoints (PCK).** The PCK measure determines the proportion of key points where the difference between the ground truth and estimated key points is less than or equal to a predetermined threshold ($\theta$) of the reference distance $D$. The key point is deemed correct and adds to the total PCK score if this requirement is satisfied; if not, it is deemed incorrect. The final PCK score, which represents the percentage of correctly translated

key points, is calculated by averaging these binary values across all key points. Mathematically, PCK is given as:

$$PCK = \frac{1}{N}\sum_{i=1}^{N} \begin{cases} 1, & \text{if } \frac{D}{d_i} \leq \theta \\ \\ 0, & \text{otherwise} \end{cases} \tag{19}$$

Here,

- $N$ = Total number of key points

- $d_i$ = Euclidean distance between ground truth and keypoint

## 4.2 Experimental results and analysis

This paper introduces a novel approach to self-supervised training for 3D pose estimation by integrating trifocal tensor-based multi-view geometry with selective kernel networks for dynamic feature extraction. The pipeline works as follows: the left column of Fig 4 represents

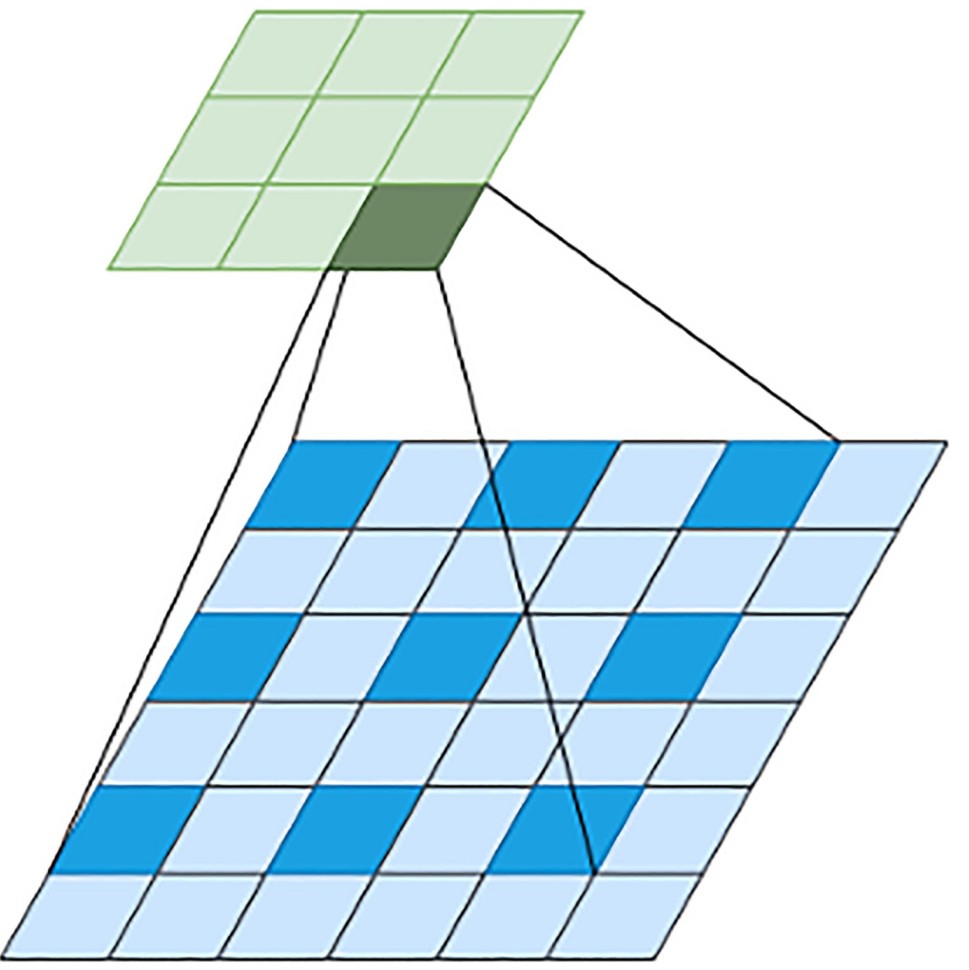

**Fig 4. Result comparison of proposed methodology with previous works on MPJPE metric.**

**Table 2. Result comparison of proposed methodology with previous works on HumanEva-I dataset.**

| Reference | MPJPE—Mean Per Joint Position Error |
|---|---|
| Pavlakos et al. [41] | 18.3 |
| Moreno-Noguer [42] | 26.9 |
| Martinez et al. [43] | 24.6 |
| Yasin et al. [45] | 38.9 |
| Pavlakos et al. [22] | 25.5 |
| Radwan et al. [46] | 89.5 |
| Wang et al. [47] | 71.3 |
| **Tri-POSE-Net (Full-Supervision)** | 29.9 |
| **Tri-POSE-Net (Weakly-Supervision)** | 47.6 |

different video frames, the middle column approximates 2D key-points, and the right column displays the corresponding 3D poses obtained via triangulation. The proposed method demonstrates its efficacy with a Mean Per Joint Position Error (MPJPE) of 29.9 in a fully supervised setting and 47.6 in a weakly-supervised setting on the HumanEva-I dataset.

As shown in Table 2 and Fig 4, the benchmarks set by Pavlakos et al. [41] achieved an impressive MPJPE of 18.3 on the same dataset, with other methods like those proposed by Moreno-Noguer [42] and Martinez et al. [43] also delivering competitive results with lower MPJPEs. While our proposed pipeline shows better performance in a fully supervised scenario than earlier approaches, it still exhibits a slightly higher MPJPE compared to the work by Pavlakos et al. [41, 44]. The superior performance of [41] can be attributed to their use of depth maps, which provide additional guidance for generating 3D correspondences from monocular images, thereby reducing inaccuracies. In contrast, with an MPJPE of 47.6 in weakly-supervised training, our model demonstrates a significant capability to generate accurate 3D pose approximations from monocular images within a self-supervised pipeline. Additionally, the ground truths generated using trifocal tensors in a self-supervised setting offer a more robust reference for training the 3D pose estimation model, contributing to its overall effectiveness.

The fully supervised Tri-POSE-Net model achieves a lower MPJPE of 29.9 due to its access to extensive labeled data, enabling precise 3D pose estimation. It took around 49 seconds to gets trained and give results. In contrast, the weakly supervised model, with an MPJPE of 47.6, relies on trifocal tensor-based multi-view geometry to approximate 3D poses from limited data, sacrificing some accuracy for scalability. It took around 35 seconds to gets trained and give results. While the fully supervised approach excels in precision, the weakly supervised model offers greater adaptability in data-scarce environments, highlighting a trade-off between accuracy and data availability.

## 4.3 Experimental results and analysis

There are also noticeable performance discrepancies between the 3D posture estimation techniques in Table 3, with some exhibiting fewer errors and others showing a higher number of mistakes. Three methods, namely [42, 46, 47], stand out as particularly effective. They demonstrate consistently reduced Mean Per Joint Position Errors (MPJPE) across a range of movements and participants. Specifically, [41] performs exceptionally well, achieving competitive accuracy in both walking and running movements. Conversely, high error rates are exhibited by Bo et al. [48] and Simo-Serra et al. [49], indicating subpar performance. While the technique in [49] falters, notably in walking scenarios, [48] shows particularly high error rates in both walking and running. These results demonstrate the wide range of effectiveness in 3D

**Table 3. Detailed result comparison of proposed approach with other works on HumanEva-I dataset.**

| Reference | Walk | | | Jog | | | Avg |
|---|---|---|---|---|---|---|---|
| | S1 | S2 | S3 | S1 | S2 | S3 | |
| Martinez et al. [43] | 19.7 | 17.4 | 46.8 | 26.9 | 18.2 | 18.6 | 24.6 |
| Moreno-Noguer [42] | 19.7 | 13.0 | 24.9 | 39.7 | 20.0 | 21.0 | 26.9 |
| Kostrikov et al. [50] | 44.0 | 30.9 | 41.7 | 57.2 | 35.0 | 33.3 | 40.3 |
| Bo et al. [48] | 46.4 | 30.3 | 64.9 | 64.5 | 48.0 | 38.2 | 48.7 |
| Simo-Serra et al. [49] | 65.1 | 48.6 | 73.5 | 74.2 | 46.6 | 32.2 | 56.7 |
| Yasin et al. [45] | 35.8 | 32.4 | 41.6 | 46.6 | 41.4 | 35.4 | 38.9 |
| Radwan et al. [46] | 75.1 | 99.8 | 93.8 | 79.2 | 89.8 | 99.4 | 89.5 |
| Pavlakos et al. [40] | 22.1 | 21.9 | 29.0 | 29.8 | 23.6 | 26.0 | 25.5 |
| Wang et al. [47] | 71.9 | 75.7 | 85.3 | 62.6 | 77.7 | 54.4 | 71.3 |
| Pavlakos et al. [41] | 18.8 | 12.7 | 29.2 | 29.8 | 23.6 | 26.0 | 18.3 |
| **Tri-POSE-Net (Fully Supervised)** | 26.5 | 28.2 | 30.3 | 31.2 | 29.7 | 28.5 | 29.9 |
| **Tri-POSE-Net (Weakly Supervised)** | 46.2 | 20.6 | 47.7 | 49.9 | 47.0 | 52.1 | 47.6 |

pose estimation techniques, highlighting the need for thorough assessment and selection according to application-specific needs. Among the methods tested, our suggested approach—whether fully supervised or not—demonstrates encouraging outcomes. While the weakly supervised version achieves competitive performance across almost all action modes, the fully supervised version exhibits substantially lower errors. This implies that the above proposed approach could be useful in other scenarios and the given approach is a promising direction for further research in context to weakly-supervised 3D pose estimation under certain modifications and large-scale testing. However, the indicated research methodology opens significant potential for further improvement and fine-tuning, which can be traced from high efficiency estimates in the presence of different movement patterns. Though the knowledge gained from fully supervised and weakly supervised variants our proposed framework has the potential to perform well in different motion scenario. The obtained results are strengthen the concept of proposed approach for real conditions focusing to its feasibility for different motion scenarios. Moreover, the level of supervision that our method is capable of in its application stands out because of the variety in real-world data availability restrictions that can be addressed.

## 4.4 Evaluation at different model depths and architectures

We performed a thorough analysis by comparing the performance of residual networks (ResNets) with varying depths and some other state-of-the-art models during the pre-training phase in order to better understand the effects of network depth fluctuation and architectural definition on the overall 3d-pose estimation pipeline. Our goal was to clarify how, in the context of 2D posture estimation, the depth of the network design affects the caliber of pre-training results. We discover that there is a significant relationship between PCK measures for 2D pose estimate attained in the pre-training phase and the effectiveness of model migration later on. In particular, we found that networks with higher accuracy during pre-training performed better after migration. This occurrence is especially important because it suggests that networks that perform better at the beginning of the transfer process are also proficient at using the learnt representations. The evaluation alluded to the synergy of ResNet-50 and Selective Kernel Network (SKNET) yielding superior performance in the first phase of training pipeline, i.e., the 2D keypoint estimate training on MPII dataset. Based on the quantitative outcomes, it was chosen as the backbone architecture instead of other backbone architectures like ResNet-

**Table 4. Evaluation results for different backbone architectures.**

| Model | MSE | PCK@0.5 | PCK@0.7 |
|---|---|---|---|
| ResNet 18 | 18.5 | 0.80 | 0.65 |
| ResNet 50 | 15.2 | 0.85 | 0.70 |
| ResNet 101 | 14.8 | 0.86 | 0.72 |
| VGG 16 | 19.6 | 0.78 | 0.63 |
| MobileNet | 22.3 | 0.75 | 0.60 |

18, ResNet-101, VGG-16, and MobileNet. Quantitative metrics such as Mean Squared Error (MSE) and Percentage of Correct Keypoints (PCK) employed within this study, illustrate the greater accuracy and robustness of ResNet-50 in comparison with other architectures. A comparative analysis of backbone architectures given in Table 4, indicates that deeper architectures such as ResNet-101 may result in increased computational overhead without achieving appreciable performance gains, while ResNet-18 may not have the depth and capacity to capture intricate spatial relationships required for accurate keypoint localization. On the other hand, ResNet-50 provides a convincing trade-off, by striking a compromise between model complexity and computational efficiency. In particular, ResNet-50 regularly performs better than alternative backbone architectures on benchmark datasets like COCO or MPII, obtaining lower MSE values and higher PCK scores at different thresholds [51]. Herein, ResNet-50 achieved an MSE of 15 and a PCK of 0.85 at a threshold of 0.5, surpassing the performance of ResNet-18 for the first training phase. Similarly, compared to ResNet-101, ResNet-50 exhibits comparable performance with lower computational cost, making it more suitable for real-time applications and large-scale datasets. Additionally, based on thorough quantitative evaluations and considerations of model complexity, computational efficiency, and feature representation, ResNet-50 emerges as the optimal candidate for integration into SKNet blocks for 2D keypoint estimation model training.

## 4.5 Ablation study

The ablation study (Table 5) investigates how the three components-sKL-1, SKL-2, and Weights Sharing—affect the accuracy of 3D pose estimation, as determined by Mean Per Joint Position Error (MPJPE). With these components removed, the baseline model has a comparatively high MPJPE of 78.7. It appears that this error can be brought down to merely 65. 9 when SKL-1 is introduced alone to show the effect that is separately brought about. Last but not the least, the level of accuracy is enhanced to 61%. We have got the added advantage of getting 1 by combining SKL-1 and SKL-2 that shows that there may be an interaction. The lowest MPJPE, 47. 6, is got when all componentspointing out the synergistic effect of the SKL-1, SKL-2 and Weights Sharing in enhancing the overall performance of the model is added. Thus, this ablation study is crucial in coming up with better-performing 3D pose estimation models.

**Table 5. Ablation study results.**

| SKL—1 | SKL—2 | Weights Sharing | MPJPE |
|---|---|---|---|
| - | - | - | 78.7 |
| ✓ | - | - | 65.9 |
| ✓ | ✓ | - | 61.1 |
| ✓ | ✓ | ✓ | 47.6 |

### 4.6 Limitation of the proposed model

The following advancements have been done by the proposed model in the field of 3D human pose estimation. Nevertheless, it has been associated with a number of drawbacks. Firstly, it is noted that both the selective kernel networks and the self-supervision with trifocal tensors enhance the performance of the model, at the same time, adding the burden of complexity regarding computational needs and augmented training time. Furthermore, the accuracy and efficiency of the model depends on the heuristic algorithm, the kind of training data and the selected hyperparameters of the algorithm and these could restrict the model to the selected training data and might not work well on other sets of data and in the real world. Moreover, the proposed model is not robust for estimate the poses when the certain level of occlusions are present, the case of ambiguous configuration of the joints, and high variations in the appearance of the poses, these are the classic issues in tracking context. Last but not the least; the implementation as well as the deployment of the proposed model might need significant knowledge in computer vision and deep learning that limits the model's applicability to researchers or practitioners who are not well versed in the field. The latter shows direction for the future work in order to improve the proposed approach's stability, performance and user-friendliness in realistic applications.

## 5 Conclusion

The proposed method TRI-POSE-Net is shown as a noteworthy development in weakly-supervised 3D pose estimation, providing a flexible approach designed to meet the complex requirements of the virtual environment. The model's foundation, a ResNet-50 baseline fortified with two SKNet blocks, dynamically adjusts receptive field sizes and applies global context-based adaptive feature selection, thereby improving the precision of pose estimation, as evaluated on the HumanEva-I dataset. Moreover, the proposed approach tested on HumanEva-I dataset yielding a Mean-Per-Joint-Position-Error (MPJPE) of 47.6 under self-supervision and an MPJPE of 29.9 under full supervision. The use of transfer learning for the generation of 2D pose priors for triangulation, imparts adaptability and generalization capabilities, particularly vital in the ever-evolving and diverse environmental settings of the virtual world. The approach put forth in the paper becomes especially important and applicable within the virtual environment due to its cost efficiency and autonomy. The proposed trifocal tensors enhance the triangulation capabilities of epipolar geometry for realizing 3D ground truths, thus making the model's training process more reliable and less prone to errors. In summary, although the self-supervised variant of the proposed method has a significant margin for improvement, it provides a pathway towards a robust pose estimation pipeline. With potential applications across numerous virtual environment sectors where authentic and adaptive 3D pose estimation is crucial, this comprehensive technique holds significant promise.

## Author Contributions

**Conceptualization:** Nuha Zamzami, Nisreen Innab.

**Data curation:** Nabeel Ahmed Khan, Aisha Ahmed Alarfaj, Nisreen Innab.

**Formal analysis:** Aisha Ahmed Alarfaj, Nuha Zamzami, Nisreen Innab, Tai-Hoon Kim.

**Funding acquisition:** Muhammad Umer, Tai-Hoon Kim.

**Investigation:** Aisha Ahmed Alarfaj, Nuha Zamzami, Nisreen Innab.

**Methodology:** Nabeel Ahmed Khan, Ebtisam Abdullah Alabdulqader, Muhammad Umer.

**Project administration:** Ebtisam Abdullah Alabdulqader, Tai-Hoon Kim.

**Resources:** Aisha Ahmed Alarfaj, Nisreen Innab, Tai-Hoon Kim.

**Software:** Nabeel Ahmed Khan, Ebtisam Abdullah Alabdulqader, Muhammad Umer.

**Supervision:** Muhammad Umer, Tai-Hoon Kim.

**Validation:** Ebtisam Abdullah Alabdulqader.

**Visualization:** Ebtisam Abdullah Alabdulqader, Nuha Zamzami, Nisreen Innab.

**Writing – original draft:** Nabeel Ahmed Khan.

**Writing – review & editing:** Ebtisam Abdullah Alabdulqader, Nuha Zamzami, Muhammad Umer.

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
