## [Decision Letter · Decision Letter 0]

19 Apr 2024

PONE-D-24-10615TRI-POSE-Net: Adaptive 3D Pose Estimation in the Metaverse through Selective Kernel Networks and Self-Supervision with Trifocal TensorsPLOS ONE

Dear Dr. Umer,

Thank you for submitting your manuscript to PLOS ONE. After careful consideration, we feel that it has merit but does not fully meet PLOS ONE’s publication criteria as it currently stands. Therefore, we invite you to submit a revised version of the manuscript that addresses the points raised during the review process.

We look forward to receiving your revised manuscript.

Kind regards,

Nouman Ali

Academic Editor

PLOS ONE

Journal Requirements:

3. In the online submission form you indicate that your data is not available for proprietary reasons and have provided a contact point for accessing this data. Please note that your current contact point is a co-author on this manuscript. According to our Data Policy, the contact point must not be an author on the manuscript and must be an institutional contact, ideally not an individual. Please revise your data statement to a non-author institutional point of contact, such as a data access or ethics committee, and send this to us via return email. Please also include contact information for the third party organization, and please include the full citation of where the data can be found.

4. Please be informed that funding information should not appear in the Acknowledgments section or other areas of your manuscript. We will only publish funding information present in the Funding Statement section of the online submission form. Please remove any funding-related text from the manuscript.

5. We note that Figure 5 includes an image of a [patient / participant / in the study].

Reviewers' comments:

Reviewer's Responses to Questions

**Comments to the Author**

1. Is the manuscript technically sound, and do the data support the conclusions?

Reviewer #1: Yes

Reviewer #2: Yes

2. Has the statistical analysis been performed appropriately and rigorously? 

Reviewer #1: Yes

Reviewer #2: No

3. Have the authors made all data underlying the findings in their manuscript fully available?

Reviewer #1: No

Reviewer #2: Yes

4. Is the manuscript presented in an intelligible fashion and written in standard English?

Reviewer #1: Yes

Reviewer #2: No

5. Review Comments to the Author

Reviewer #1: The authors present both a new neural network architecture as well as a method to recover groundtruth 3D human pose estimation from multiple images, that can be used to supervise a network predicting 3D joint positions.

The neural network architecture is a combination of ResNet-50 and Selective Kernel Networks blocks. The 3D pose annotation is recovered using trifocal tensors, solving a projective geometry estimation problem between the image-space joint detections from three different images with known camera poses.

The two aspects of the contribution (architecture and self-supervision) have little to do with each other, and neither are properly evaluated on their own. If the central point is the architecture, the choices should be better justified and evaluated, for example looking at the impact of SKNet blocks on different backbones, and looking at the results on different datasets. If the self-supervision is the central aspect, its benefits should be verified on different models, and more in-depth analysis of the quality of the generated groundtruth would be beneficial. It would also be interesting to look at whether that supervision improves upon the 2d joint position prediction once reprojected, or if the results are only as good as (or worse than) the 2d joint position estimation network used to create the annotation.

The method presented also doesn't outperform comparable state-of-the-art (3 reported methods outperform it, despite it using an additional dataset as pre-training)

The reference presented do not always include all of the relevant informations, especially the conference venues. Many of them also refer to Github repository. No reference to the paper introducing ResNet or SKNet is done in the introduction despite discussing it.

While the problem of fast and accurate Human Pose Estimation is relevant and of interest to this venue, nothing in the method presented relate it specifically to the "metaverse" (which is not a well-defined technical term in the first place), so putting it in the title seems unnecessary.

As a small point, some discussion regarding the required accuracy for the camera pose of the three view used would be a welcome addition.

The paper also has typos or smaller mistakes (including sentences that are hard to parse or incorrect), here is a subset:

- the title should state Human Pose Estimation instead of Pose Estimation

- restraints instead of constraints

- enlisted instead of listed

- parametric sharing instead of parameter sharing

- lineament is not clear, possibly misused

- N and A used in 3.1 before being defined

- Aec and Bec used without being defined in equation 5

- the text of Figure 3 and 7 is a little too small

- In figure 8, the keypoints are not visible

Reviewer #2: Accurate and flexible 3D pose estimation for virtual entities is a strenuous task in both computer vision and the metaverse. Conventional methods struggle to capture realistic movements; thus, creative solutions that can handle the complexities of genuine avatar interactions in dynamic virtual environments are imperative. In order to tackle the problem of precise 3D pose estimation, this work introduces TRI-POSE-Net, a model intended for scenarios with limited supervision. The proposed technique, which is based on ResNet-50 and includes integrated Selective Kernel Network (SKNet) blocks, has proven to be efficient for feature extraction customised specifically to pose estimation scenarios. Furthermore, trifocal tensors and their trio-view geometry allow us to generate 3D ground truth poses from 2D poses, resulting in more refined triangulations. Through the proposed approach, the 3D poses can be estimated from a single 2D RGB image. Moreover, the proposed approach was evaluated on the HumanEva-I dataset yielding a Mean-Per-Joint-Position-Error (MPJPE) of 47.6 under self-supervision and an MPJPE of 29.9 under full supervision. In comparison with the other works, the proposed work has performed well in the self-supervision paradigm. The paper has potentials but it requires some major revisions before making any decision on it.

1. Why the authors forcefully adding the terms of meta-verse in the paper? I would suggest if author remove the terms of metaverse from the paper.

2. Add the results of the proposed model in the abstract.

3. Add some potential references in the paper replacing too much conference and website links from the paper.

4. Add some details (single paragraph) about related work section before moving to section 2.1 directly.

5. In section 2.2, references 11-13 and 14-17 are not cited properly, try to discuss them separately or remove less relevant papers.

6. There are too many short forms used in the paper. Try to add abbreviation table to improve the readability of the paper.

7. There is no graph to visualize the results. Remembering values is difficult.

8. Explain how MPJPE is best evaluation parameter for this model.

9. Re-write conclusion section by adding result values.

10. What are the limitations of the proposed model?

11. English editing is necessary, there are many typos and grammatical mistake in the paper. Authors are advised to revise the paper critically.

6. PLOS authors have the option to publish the peer review history of their article (what does this mean?). If published, this will include your full peer review and any attached files.

Reviewer #1: **Yes: **Mehmood Ashraf

Reviewer #2: No

---

## [Author Response · Author response to Decision Letter 0]

15 Jun 2024

We have provided separate response to reviewers PDF file.

---

## [Decision Letter · Decision Letter 1]

20 Aug 2024

PONE-D-24-10615R1TRI-POSE-Net: Adaptive 3D Human Pose Estimation Through Selective Kernel Networks and Self-Supervision with Trifocal TensorsPLOS ONE

Dear Dr. Umer,

Thank you for submitting your manuscript to PLOS ONE. After careful consideration, we feel that it has merit but does not fully meet PLOS ONE’s publication criteria as it currently stands. Therefore, we invite you to submit a revised version of the manuscript that addresses the points raised during the review process.

We look forward to receiving your revised manuscript.

Kind regards,

Nouman Ali

Academic Editor

PLOS ONE

Reviewers' comments:

Reviewer's Responses to Questions

**Comments to the Author**

1. If the authors have adequately addressed your comments raised in a previous round of review and you feel that this manuscript is now acceptable for publication, you may indicate that here to bypass the “Comments to the Author” section, enter your conflict of interest statement in the “Confidential to Editor” section, and submit your "Accept" recommendation.

Reviewer #1: All comments have been addressed

Reviewer #2: (No Response)

Reviewer #3: (No Response)

Reviewer #4: (No Response)

2. Is the manuscript technically sound, and do the data support the conclusions?

Reviewer #1: Yes

Reviewer #2: Yes

Reviewer #3: Yes

Reviewer #4: Yes

3. Has the statistical analysis been performed appropriately and rigorously? 

Reviewer #1: Yes

Reviewer #2: Yes

Reviewer #3: Yes

Reviewer #4: Yes

4. Have the authors made all data underlying the findings in their manuscript fully available?

Reviewer #1: Yes

Reviewer #2: Yes

Reviewer #3: (No Response)

Reviewer #4: Yes

5. Is the manuscript presented in an intelligible fashion and written in standard English?

Reviewer #1: Yes

Reviewer #2: Yes

Reviewer #3: Yes

Reviewer #4: Yes

6. Review Comments to the Author

Reviewer #1: Authors did well to improve this paper and to address my comments. The revised version of the paper looks okay for me to get accepted in PLOS ONE journal.

Reviewer #2: Author have addressed my all main queries and the quality of this manuscript has been improved. Here are my few minor queries:

1. How authors have selected the research that has been used for comparison, an explanation is required.

2. The values presented in graphs and tables require more discussion about results.

3. More discussions are required about Fully Supervised and Weakly Supervised model results difference.

Reviewer #3: Accurate and flexible 3D pose estimation for virtual entities is a strenuous task in computer vision applications. Conventional methods struggle to capture realistic movements; thus, creative solutions that can handle the complexities of genuine avatar interactions in dynamic virtual environments are imperative. In order to tackle the problem of precise 3D pose estimation, this work introduces TRI-POSE-Net, a model intended for scenarios with limited supervision. The proposed technique, which is based on ResNet-50 and includes integrated Selective Kernel Network (SKNet) blocks, has proven to be efficient for feature extraction customised specifically to pose estimation scenarios. Furthermore, trifocal tensors and their trio-view geometry allow us to generate 3D ground truth poses from 2D poses, resulting in more refined triangulations. Through the proposed approach, the 3D poses can be estimated from a single 2D RGB image. Moreover, the proposed approach was evaluated on the HumanEva-I dataset yielding a Mean-Per-Joint-Position-Error (MPJPE) of 47.6 under self-supervision and an MPJPE of 29.9 under full supervision. In comparison with the other works, the proposed work has performed well in the self-supervision paradigm. This manuscript has already been reviewed once and the quality of the manuscript has been improved by the authors. Here are my queries

1. Claim 3, <<this 3d="" enhance="" of="" paper="" pose="" process="" proposes="" tensors="" the="" to="" triangulation="" trifocal="" use=""> correspondences between three views. Offering better capturing of spatial

relationships then the fundamental matrix>>, here clarification is required that this is a sort of global spatial information or local?

2. There are many headings without text such as 4 and 4.1. there must be some text associated with each heading

3. A clarification is required about the selection of parameters and performance evaluation criteria

4. Runtime analysis should be presented .</this>

Reviewer #4: I have carefully analyzed the first version and revised version of the paper. The paper improved a lot after the first round of reviewers with the help of reviewers' comments. Still, I have some minor concerns on the paper that need to be addressed to further improve the paper.

1. Please add some more lines to justify the name "Tri-Pose-Net"

2. The addition of the latest research works of 2023-24 on 3d pose estimation needed to be added to the paper.

3. Stick to 1 term either Tri-POSE-Net, Tri-Pose-Net or Tri-Pose Net in the paper.

4. In Table 1, write Tri-Pose Net instead of the proposed method.

5. In Table 2, write Tri-Pose Net instead of the proposed method.

6. Try to define the acronym of words at first usage and then use that acronym throughout the paper.

7. PLOS authors have the option to publish the peer review history of their article (what does this mean?). If published, this will include your full peer review and any attached files.

Reviewer #1: No

Reviewer #2: No

Reviewer #3: No

Reviewer #4: No

---

## [Author Response · Author response to Decision Letter 1]

23 Aug 2024

We have provided a separate PDF to address reviewers concern.

---

## [Decision Letter · Decision Letter 2]

8 Sep 2024

TRI-POSE-Net: Adaptive 3D Human Pose Estimation Through Selective Kernel Networks and Self-Supervision with Trifocal Tensors

PONE-D-24-10615R2

Dear Dr. Umer,

We’re pleased to inform you that your manuscript has been judged scientifically suitable for publication and will be formally accepted for publication once it meets all outstanding technical requirements.

Kind regards,

Nouman Ali

Academic Editor

PLOS ONE

Additional Editor Comments (optional):

Reviewers' comments:

Reviewer's Responses to Questions

**Comments to the Author**

1. If the authors have adequately addressed your comments raised in a previous round of review and you feel that this manuscript is now acceptable for publication, you may indicate that here to bypass the “Comments to the Author” section, enter your conflict of interest statement in the “Confidential to Editor” section, and submit your "Accept" recommendation.

Reviewer #2: All comments have been addressed

Reviewer #3: All comments have been addressed

Reviewer #4: All comments have been addressed

2. Is the manuscript technically sound, and do the data support the conclusions?

Reviewer #2: Yes

Reviewer #3: Yes

Reviewer #4: Partly

3. Has the statistical analysis been performed appropriately and rigorously? 

Reviewer #2: Yes

Reviewer #3: Yes

Reviewer #4: Yes

4. Have the authors made all data underlying the findings in their manuscript fully available?

Reviewer #2: Yes

Reviewer #3: Yes

Reviewer #4: Yes

5. Is the manuscript presented in an intelligible fashion and written in standard English?

Reviewer #2: Yes

Reviewer #3: Yes

Reviewer #4: Yes

6. Review Comments to the Author

Reviewer #2: Manuscript is improved now, and author have addressed my all queries and suggestion. Therefore, i am recommending acceptance of this manuscript.

Reviewer #3: This paper is now in much improved form and is Recommended for acceptance. My all queries have been addressed

Reviewer #4: All comments are addressed . Everything is good to go . Author addresses all basic comments as well

7. PLOS authors have the option to publish the peer review history of their article (what does this mean?). If published, this will include your full peer review and any attached files.

Reviewer #2: No

Reviewer #3: No

Reviewer #4: No

---

## [Editor Report · Acceptance letter]

11 Oct 2024

PONE-D-24-10615R2 

PLOS ONE

Dear Dr. Umer, 

I'm pleased to inform you that your manuscript has been deemed suitable for publication in PLOS ONE. Congratulations! Your manuscript is now being handed over to our production team.

Kind regards, 

on behalf of

Dr. Nouman Ali 

Academic Editor

PLOS ONE